# COVID-19 pandemic hits differently: examining its consequences for women's livelihoods and healthcare access – a cross-sectional study in Kinshasa DRC

Falone Nkweleko Fankam ®,[1] William Ugarte,[2] Pierre Akilimali,[3] Junior Ewane Etah,[4] Eva Åkerman[5]

For numbered affiliations see end of article.

**Correspondence to**
Eva Åkerman;
eva.akerman.2@ki.se

## ABSTRACT

**Objectives** The emergence of the COVID-19 pandemic led to multiple preventive actions as primary interventions to contain the spread of the virus. Globally, countries are facing enormous challenges with consequences for use of social, economic and health services. The Democratic Republic of Congo (DRC) was among the African countries implementing strict lockdown at the start of the pandemic, resulting in shortages and decreased access to services. The adverse effects of the pandemic had unpleasant consequences for the country. This study aimed to examine the association between COVID-19 pandemic-related factors, sociodemographic factors, and the need to visit healthcare facilities, including family planning services, among women aged 15–49 years in the DRC.

**Methods** We conducted a secondary analysis of a performance monitoring for action (PMA) cross-sectional COVID-19 phone survey in Kinshasa, DRC, which had a response rate of 74.7%. In total, 1325 randomly selected women aged 15–49 years from the Kinshasa province who had previously participated in the PMA baseline survey participated in the survey. Bivariate and multivariate logistic regressions were used to assess associations.

**Results** The COVID-19 pandemic and related factors affected 92% of women in the Kinshasa province socioeconomically. A majority were highly economically dependent on their partner or some other sources for their basic needs to be met, and even more worried about the future impact of the pandemic on their household finances. Over 50% of women did not attempt visiting a health service, with some of the top reasons being fear of being infected with COVID-19 and not being able to afford services. We found a significant association between age groups and contraceptive use. The need for and use of contraceptives was higher among women aged 25–34 years than those aged 15–24 or 35–49 years.

**Conclusion** Effective social/economic support to women and girls during pandemics and in crises is essential as it can have lasting beneficial effects on many domains of their lives, including their ability to access health services and the contraceptives of their choice.

## STRENGTHS AND LIMITATIONS OF THIS STUDY

⇒ The study used a cross-sectional design, providing a snapshot of the situation during the COVID-19 pandemic in Kinshasa, Democratic Republic of Congo.
⇒ The study involved a large sample of 1325 women residing in Kinshasa with good response rate (74.7%).
⇒ Data were collected using phone surveys, which minimised the risk of infection and enabled efficient data collection.
⇒ One limitation of this study is the lack of causal conclusions due to the cross-sectional design.
⇒ The sample is limited to women in Kinshasa, which is an urban area and limiting generalisability in rural areas.

## INTRODUCTION

The emergence of the COVID-19 pandemic resulted in multiple restrictive actions being taken as primary preventive interventions in most countries.[1] Globally, many countries were remarkably unprepared, being far from ready to cope with the huge increase in critically ill persons. A scoping review by Bolarinwa *et al*[2] including studies from both high-income and low-middle-income countries reported the severe impact of COVID-19 on family planning (FP) services, sexual behaviour, and maternal and child health services because of limited access to the preferred choices, especially for those with FP needs. Globally, the impact of the pandemic meant that sexual and reproductive health (SRH) services and support ranging from condoms, contraceptives, HIV testing and treatment, and uninterrupted hormone treatment for transgender persons might be limited, if available at all.[3] Thus, the pandemic has been a threat to several countries, stretching health systems to their limits, especially those with preexisting

crises and humanitarian conditions—as is the case in the Democratic Republic of Congo (DRC).[4] Thus, COVID-19 intensifies pre-existing health hardships, for instance, the DRC had undergone several disease outbreaks in the last 2 years, including measles, cholera and Ebola.[4] The most recent Ebola outbreak was considered the second largest in the world[5] and claimed thousands of lives.[5] The long-lasting Ebola epidemic, combined with insecurity and armed conflicts, has caused an inefficient and insufficient supply of SRH services, particularly for women and young people.[6] In addition, lockdowns and resources allocated to the COVID-19 response further compounded these problems by limiting access to essential SRH services among women and girls.[6]

Furthermore, the first lockdown after the first COVID-19 case was discovered in Kinshasa on 10 March 2020 included a suspension of flights from COVID-19-infected countries and a movement prohibition within the country, further limiting access to services[7] and possibly obstructing the supply chain and increasing shortages for those who could access services. Also, people were frustrated and devasted by severe social and economic hurdles, which led to intensified struggles among women and girls.[8] A recent publication showed an increased need for contraception among nulliparous women in the DRC during the COVID-19 pandemic.[9] No such need was observed among women who already had children.

Setting aside resources and providing equal access to essential healthcare services, including FP services, during crises and pandemics is crucial. Various health and government institutions promoted a range of measures to contain the pandemic that focused on individual behaviours, with little attention to social, economic, contextual[10] or psychological factors.[11 12] Thus, this research was aimed at better understanding the social and economic consequences of the pandemic relative to the need for health services, including FP, among women in the urban setting of the DRC context. Recognising these factors and consequences would potentially help stakeholders strengthen their preparation, budgeting and delivery of SRH services globally and gain a better grasp of possible short-term and long-term consequences of pandemic lockdowns on women.

The study aimed to investigate if there was an association between COVID-19 pandemic-related factors, socio-demographic factors and the need for visiting health facilities, including FP services, in women aged 15–49 years in the DRC.

## METHODS
### Study design, population and sample size
The study used a cross-sectional design and was conducted by phone between May and June 2020. Participants were women aged 15–49 years randomly selected from households in the Kinshasa province, who participated in the Performance Monitoring for Action (PMA) baseline survey between December 2019 and February 2020.[12]

Overall, 69.5% of the baseline participants consented to take part in future studies and provided a phone number. Of the 1773 eligible participants, 77.6% (n=1375) were reached and 96.3% (n=1325) of them completed the phone survey for a final response rate of 74.7% among eligible women.[13] Stata survey procedures were used to account for the sampling design and selection weights.

### Patient and public involvement
This secondary data analysis study involves data gathered from the PMA COVID-19 phone survey in Kinshasa. Hence, there was no patient or public involvement.

### Data collection
The survey was conducted by trained women in the designated localities using Open Data Kit (ODK). This is a publicly available software application to collect and administer surveys using mobile phones. The ODK questionnaire was adapted to restrict spontaneous skipping and thereby reduce data entry errors.[14] The data collected during the study were instantly aggregated, enabling simultaneous processing, and making it possible to make corrections and adjustments in real time, while still in the field. Throughout the data collection stage, regular monitoring was performed and feedback was given to interviewers in the event of errors, missing data or form submission problems on the central server.

### Measurements
The outcome variables for the analysis were the need to access health services, including FP and current use of contraceptives. These outcome variables were captured as: the need to visit health services, difficulties accessing health services including FP services, current contraceptive use at the time of data collection and behaviour changes in contraceptive use. Each question had a corresponding value.[15]

Accessing health services in times of COVID-19, the exposure variable, was assessed through seven closed-ended multiple choice questions for which only one answer was accepted. They related to: (A) inability to access services because of government restrictions on movement, (B) inability to afford FP services, (C) fear of being infected with COVID-19 at healthcare facilities, (D) not attempting to access healthcare services or not experiencing any need to do so, (E) having difficulties accessing transportation to reach healthcare services, (F) healthcare facility or doctor's office closed, appointment not possible and (G) preferred contraceptive method unavailable.

The sociodemographic variables included age, education, place of residence, pregnancy status, marital status, economic status and decision-making ability regarding household purchases. Economic status was based on the ability of respondents to meet their needs and those of their families or being dependent on someone else for those needs to be met. This was coded as being reliant on husband/partner, relying on oneself, relying on oneself

**Table 1** Demographic characteristics of the study population, N=1325

| Characteristics | Frequency (n) | % |
|---|---|---|
| Age group (years) | | |
| 15–24 (youth) | 474 | 35.8 |
| 25–34 (young adult) | 464 | 35.0 |
| 35–49 (adult) | 387 | 29.2 |
| Mean age (years) | 29.49±8.97 | |
| Educational level* | | |
| Primary | 44 | 3.3 |
| Secondary | 884 | 66.8 |
| Tertiary | 395 | 29.9 |
| Marital status* | | |
| Married or cohabiting | 620 | 47.0 |
| Unmarried | 703 | 53.0 |
| Currently pregnant | | |
| No | 1246 | 94.0 |
| Yes | 72 | 5.4 |
| Unsure/no response | 7 | 0.6 |

*Missing values excluded.

and husband/partner, relying on someone else or no response.

## Data analysis

Statistical analysis was done using SPSS V.26 (IBM). Descriptive statistics included means and SDs for quantitative variables while qualitative variables were summarized using frequencies and percentages. Data were checked for normality and even distribution. Cross tabulations and Pearson's $\chi^2$ test were used to compare differences in proportions between groups. Both bivariate and multivariate logistic regression analyses were used. Independent variables associated with the outcomes at $p<0.20$ in the bivariate analysis were included in the multivariate analysis. The multivariate analysis was used to control potential confounders. Logistic regression models were performed on the two dependent variables: need to access healthcare services and current use of contraceptives. The variance inflation factor (VIF) was used to test for multicollinearity between variables (predictors) in the regression model. All statistical analyses were considered significant at $p<0.05$.

## RESULTS

Table 1 presents the sociodemographic characteristics of the women included in the study. A total of 1325 women, with a mean age of 29.49±8.97 years, consented to participate in the study. Most of the women were well educated, with 66.8% having secondary education and about one-third (29.9%) having tertiary education. More than half (53.0%) of the women were unmarried. At the time of

**Table 2** Socioeconomic impact of COVID-19 restrictions on women in Kinshasa province

| Characteristics | Impact due to COVID-19 | |
|---|---|---|
| | Frequency (n) | % |
| Household income loss | | |
| None | 107 | 8.0 |
| Partial | 438 | 33.0 |
| Complete | 776 | 59.0 |
| Total | 1321* | 100 |
| Worried about the impact of COVID-19 on future household finances | | |
| Yes | 1177 | 89.8 |
| No | 133 | 10.2 |
| Total | 1310* | 100 |
| Lack of food for 24 hours | | |
| Yes | 486 | 36.8 |
| No | 834 | 63.2 |
| Total | 1320* | 100 |
| Frequency of food inadequacy (n=486) | | |
| Rarely (1–2 times) | 171 | 35.2 |
| Sometimes (3–10 times) | 239 | 49.2 |
| Often (> 10 times) | 75 | 15.4 |
| Don't know | 01 | 0.2 |
| Total | 486 | 100 |
| Currently economically reliant on husband/partner for basic needs | | |
| Yes | 392 | 29.6 |
| No | 220 | 16.6 |
| No response | 713 | 53.8 |
| Total | 1325 | 100 |
| More economically reliant on husband/partner now than before COVID-19 restrictions began | | |
| Yes | 289 | 21.8 |
| No | 102 | 7.7 |
| No response/do not know | 934 | 70.5 |
| Total | 1325 | 100 |

*Missing value excluded.

data collection, the majority (94.0%) of the women were not pregnant.

Table 2 depicts the socioeconomic impact of COVID-19 restrictions on women in the Kinshasa province. Nearly all women experienced either partial (33.0%) or complete (59.0%) loss of household income. In total, 1214 (92.0%) of women experienced household income loss due to COVID-19. Similarly, most (89.8%) of the women were worried about the impact of COVID-19 on their future household finances. In terms of food security, 36.8% of the women described having witnessed household members starving because there was not enough food to

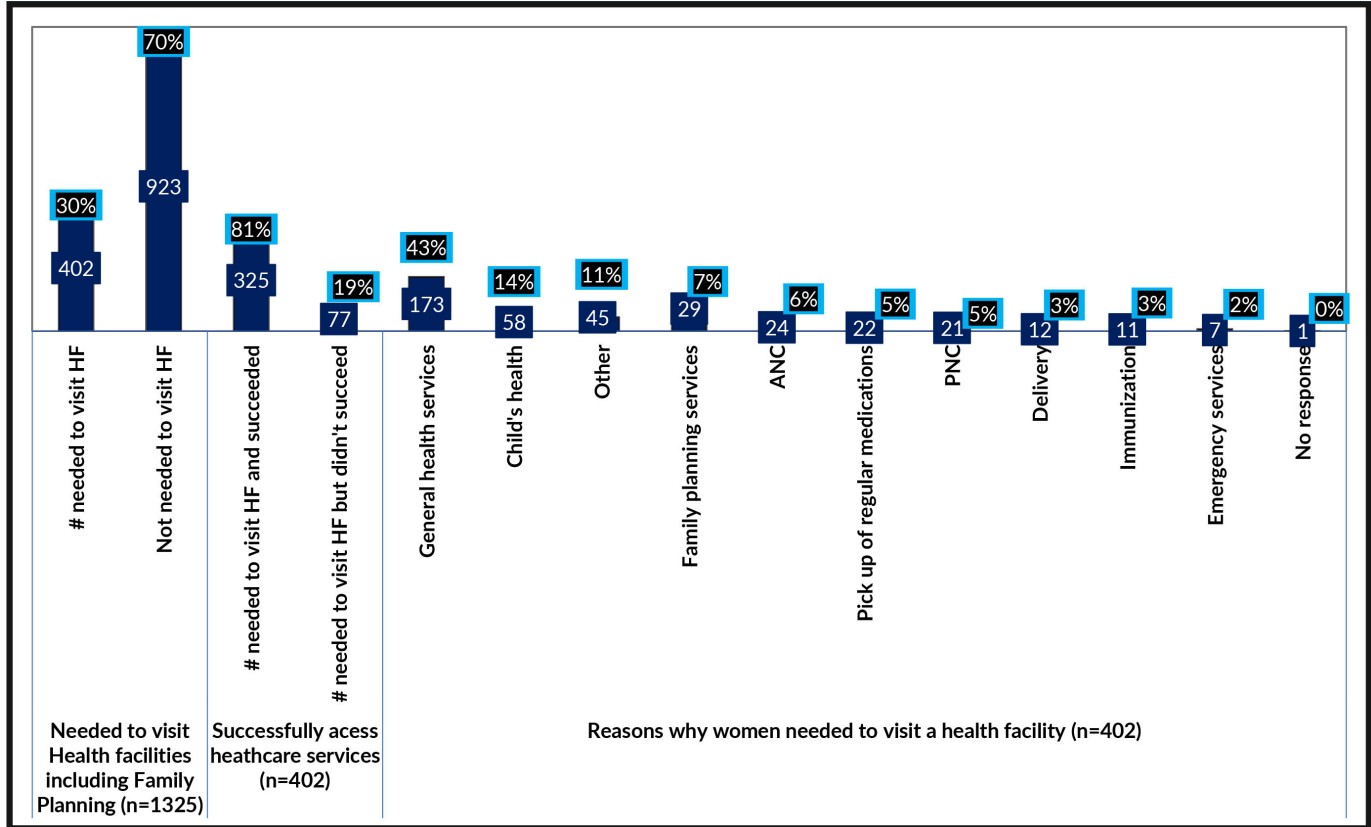

**Figure 1** Need/reasons to visit health facilities during COVID-19. ANC, antenatal care; PNC, postnatal care.

eat. The women were asked 'whether they were currently economically reliant on their husband/partner for basic needs' and 'whether they were more economically reliant on their husband/partner now than before COVID-19 restrictions began'. Less than half of the women responded to these questions. Of 612 women who responded to the first question, 392 (64.1%) stated that they were currently economically reliant on their husband/partner for basic needs. In addition, 289 (74.0%) of the 391 women had become more economically reliant on their husband/ partner than before COVID-19 restrictions began, as shown in table 2.

With regard to women's needs to visit healthcare facilities for services including FP during the COVID-19 restrictions, only 402 (30.0%) needed healthcare services. Of these 402 participants, 325 (81.0%) successfully visited facilities and had access to healthcare. The reasons for accessing healthcare were determined, ranked and presented in figure 1 (Online supplemental appendix). The largest proportion (43.0%) of the women visited health facilities for general healthcare services, though others visited the facilities due to a child's health (14.0%) or for other personal reasons (11.0%), FP services (7.0%) or antenatal care (6.0%).

Reasons for not accessing healthcare services were evaluated and presented in figure 2 (online supplemental appendix). More than half (58%) of the women reported not attempting to access healthcare services or not experiencing difficulties in accessing care. However, some of

the women did not access healthcare services for fear of being infected with COVID-19 (14.0%) while others were unable to afford healthcare services (12.0%). Other reasons included no transportation to access healthcare (6.0%), government restriction on movement (4.0%), healthcare facilities or doctors' offices closed (2.0%) and partner disapproval (1.0%).

There was a self-reported attitudinal change in the need for FP due to the COVID-19 pandemic among married or cohabiting and unmarried women, as shown in table 3. A total of 111 women stated that they had changed their mind about getting pregnant, though no statistically significant difference was observed between unmarried and married women. Most unmarried women (57.6%) responded that they would be very unhappy if they had become pregnant, which was not seen among married women (26.9%). Furthermore, a significant proportion of the unmarried women (62.0%) affirmed to wait for many years before giving birth. In line with this, more unmarried women (61.0%) than married women (38.7%) stated that they were currently using some method to delay or avoid pregnancy, as shown in table 3.

Both bivariate and multivariate analyses were conducted to assess relationships between current use of contraceptives and sociodemographic factors (see table 4). We observed that significantly more women aged 25–34 years (50.4%) were using contraceptives than those aged 15–24 years (37.1%) and 35–49 years (42.0%). A $\chi^2$ test showed a significant association

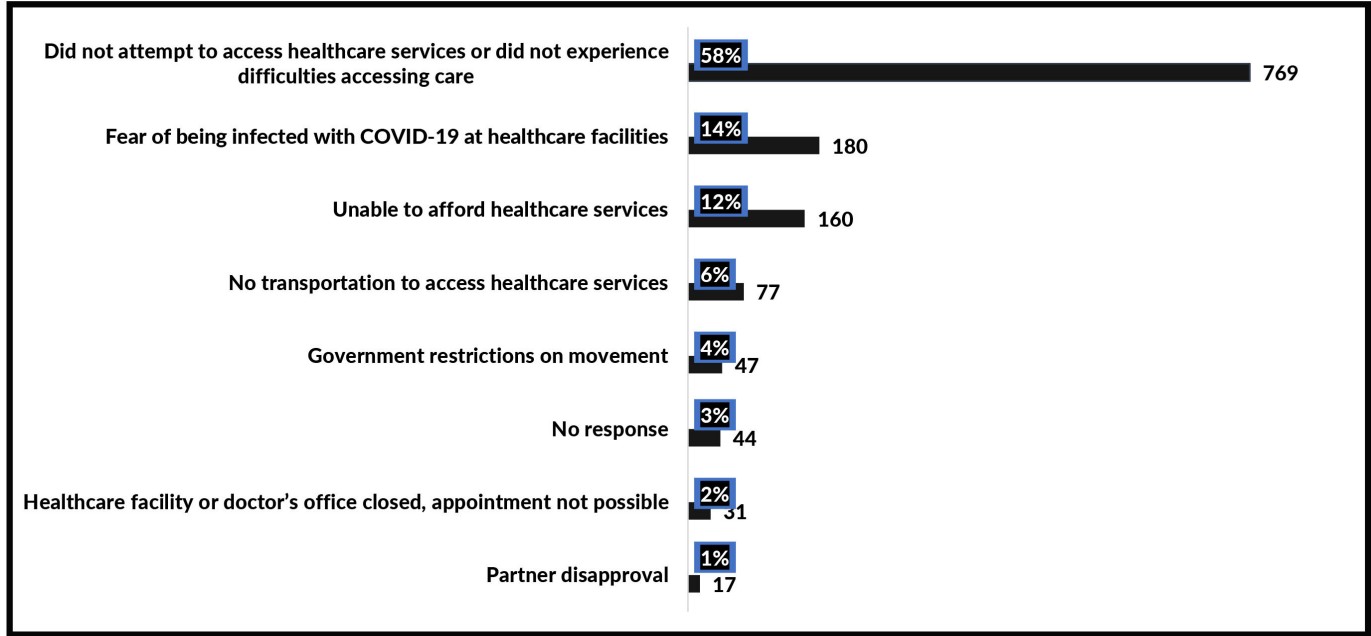

**Figure 2** Primary reasons for the inaccessibility to healthcare services among women since COVID-19 restrictions (n=1325).

between age group and current use of contraceptives. Furthermore, current use of contraceptives increased with an increase in educational level. In other words, more women with tertiary education were currently using contraceptives than those in secondary education (41.0%) and primary education (40.9%). With regard to marital status, significantly more married women (51.3%) than unmarried women (36.1%) were using contraceptives.

Though the difference was not significant, women who experienced household income loss (43.7%) used contraceptives more often than those who did not experience household income loss (39.3%). Similarly, more women who were economically reliant on their husband/partner (51.5%) reported currently using contraceptives than those not experiencing economic reliance (50.9%). After confounding in the multivariate logistic regression, being married was significantly associated (adjusted OR (AOR) 2.04

(95% CI 1.56 to 2.67, p<0.001) with current use of contraceptives. Secondary education (AOR 0.72 (95% CI 0.57 to 0.92, p=0.009) also increased the likelihood of current use of contraceptives.

To ascertain the association between sociodemographic factors and the need to access healthcare including FP, both bivariate and multivariate analyses were performed (see table 5). The bivariate analysis showed that being aged between 25 and 34 years (crude OR, COR 1.60 (95% CI 1.06 to 1.83), being in secondary education (COR 0.77 (95% CI 0.06 to 1.00), being married (COR 1.74 (95% CI 1.38 to 2.21) and having lost household income (COR 1.39 (95% CI 0.88 to 2.20) were all associated with the need to access healthcare services. However, after using the multivariate analysis, we observed that being aged 25–34 years (AOR 1.76, 95% CI 1.3 to 2.38, p<0.001) and being married (AOR 1.85 (95% CI 1.39 to 2.47, p<0.001) was significantly associated with the need to access healthcare services (see table 5).

**Table 3** Attitudinal changes due to COVID-19 restrictions, comparing married/cohabiting with unmarried women

| | | Responses | | |
| | | Married or cohabiting n=620 | Unmarried. n=703 | |
| Characteristics | Categories | frequency (%) | frequency (%) | P value |
| --- | --- | --- | --- | --- |
| Change of mind about being pregnant due to COVID-19 | Yes (n=111) | 50 (8.0) | 61 (8.7) | 0.691 |
| Would you prefer not to have any more children? | No more children (n=193) | 137 (22.1) | 56 (8.0) | <0.001* |
| Respondent's feelings if pregnant | Very unhappy (n=572) | 167 (26.9) | 405 (57.6) | <0.001* |
| Currently using method to delay or avoid getting pregnant | No (n=669) | 240 (38.7) | 429 (61.0) | <0.001* |
| Waiting time to put to birth | Years (n=651) | 215 (34.7) | 436 (62.0) | <0.001* |

For analytical purposes, married women encompassed both those legally married and those living with partners (cohabiting).
*Significant p value <0.05.

**Table 4** Bivariate and multivariate logistic regression analysis of current use of contraceptives and sociodemographic factors

| Variables | Bivariate analysis | | Multivariate analysis (n=572) | | |
| --- | --- | --- | --- | --- | --- |
| | COR (95% CI) | P value | AOR (95% CI) | P value | VIF |
| Age group (years) | | | | | 1.039 |
| 15–24 (youth) | Ref | | Ref | | |
| 25–34 (young adult) | 1.72 (1.34 to 2.24) | <0.001* | 1.22 (0.91 to 1.64) | 0.177 | |
| 35–49 (adult) | 1.23 (0.93 to 1.61) | 0.149 | 0.78 (0.56 to 1.08) | 0.136 | |
| Educational level | | | | | 1.014 |
| Tertiary | Ref | | Ref | | |
| Secondary | 0.74 (0.58 to 0.94) | 0.014* | 0.72 (0.57 to 0.92) | 0.009* | |
| Primary | 0.73 (0.39 to 1.39) | 0.350 | 0.70 (0.37 to 1.35) | 0.295 | |
| Marital status | | | | | 1.000 |
| Unmarried | Ref | | Ref | | |
| Married/cohabiting | 1.86 (1.49 to 2.32) | <0.001* | 2.04 (1.56 to 2.67) | <0.001* | |
| Economic reliance | | | | | 1.052 |
| Not reliant | Ref | | Ref | | |
| Reliant | 1.03 (0.74 to 1.43) | 0.883 | – | | |
| Household income loss | | | | | 1.001 |
| No | Ref | | Ref | | |
| Yes | 1.20 (0.80 to 1.80) | 0.379 | – | | |

*Significant p value <0.05.
AOR, Adjusted OR; CI, Confidence Interval; COR, Crude Odds Ratio; Ref, reference group; VIF, Variance Inflation Factor.

**Table 5** Bivariate and multivariate logistic regression analysis between need to access healthcare and sociodemographic factors

| Variables | Bivariate analysis | | Multivariate analysis | | Collinearity statistics |
| --- | --- | --- | --- | --- | --- |
| | COR (95% CI) | P value | AOR (95% CI) | P value | VIF |
| Age group (year) | | | | | 1.039 |
| 35–49 (adult) | Ref | | Ref | | |
| 25–34 (young adult) | 1.60 (1.06 to 1.83) | 0.002* | 1.76 (1.30 to 2.38) | <0.001* | |
| 15–24 (youth) | 0.86 (0.63 to 1.68) | 0.337 | 1.30 (0.90 to 1.88) | 0.157 | |
| Educational level | | | | | 1.014 |
| Tertiary | Ref | | Ref | | |
| Secondary | 0.77 (0.06 to 1.00) | 0.052* | 0.78 (0.60 to 1.00) | 0.056 | |
| Primary | 0.72 (0.36 to 1.45) | 0.359 | 0.71 (0.35 to 1.45) | 0.351 | |
| Marital status | | | | | 1.421 |
| Unmarried | Ref | | Ref | | |
| Married/cohabiting | 1.74 (1.38 to 2.21) | <0.001* | 1.85 (1.39 to 2.47) | <0.001* | |
| Economic reliance | | | | | 1.052 |
| Not reliant | Ref | | | | |
| Reliant | 1.07 (0.76 to 1.51) | 0.705 | – | | |
| Household income loss | | | | | 1.001 |
| No | Ref | | Ref | | |
| Yes | 1.39 (0.88 to 2.20) | 0.157 | 1.32 (0.83 to 2.11) | 0.243 | |

*Significant p value <0.05.
AOR, Adjusted OR; CI, Confidence Interval; COR, Crude Odds Ratio; Ref, reference group; VIF, Variance Inflation Factor.

## DISCUSSION

The COVID-19 pandemic has been a major global public health threat, challenging the provision and accessibility of healthcare services. The aim of this study was to investigate associations between COVID-19 pandemic-related factors, sociodemographic factors and the need for visiting health facilities including FP services in women aged 15–49 years in the Kinshasa province of DRC. According to our findings, 1 in 3 women aged 15–49 years needed to visit a health facility during the COVID-19 lockdown and 8 in 10 among them succeeded. However, findings from rural Ethiopia showed unsuccessful attempts of women trying to access health facilities during the COVID-19 lockdown, with a dramatic decline in FP accessibility.[16] Our study found that the use of healthcare services during COVID-19 restrictions was significantly affected among women. This is in line with many studies that have reported major changes in the accessibility and utilisation of healthcare services because of measures such as lockdowns and stay-at-home orders.[17–19] This differs from findings carried out in Uganda, where access to contraceptive and postnatal care remained stable during the COVID-19 pandemic[20] and in a study carried out in Kenya reporting that COVID-19 did not affect access to health services.[21]

Furthermore, we found a range of reasons for visiting health facilities, with the majority visiting for general healthcare services and some seeking FP services. Our study found consistent indications of major decreases in the utilisation of healthcare services during the COVID-19 pandemic,[22] contrary to another study[21] where there was no difference in health facility access rates during the pandemic. The reasons for women's inability to access healthcare, including FP services, in the DRC have been evaluated after COVID-19 restrictions were implemented. We observed that most women did not try to access healthcare services. Others avoided healthcare services due to fear of being infected with COVID-19, inability to afford healthcare, not having transportation to access healthcare, government restrictions on movement, closure of healthcare facilities or partner disapproval. These reasons could explain the drastic drop in the accessibility and utilisation of healthcare services among women during the COVID-19 pandemic in the province of Kinshasa. According to Rocca-Ihenacho and Alonso,[23] the population was advised not to visit a hospital unless strictly necessary; this advice seems to have applied to all citizens, including healthy pregnant women and even those with complications. Thus, the direct and indirect effects of the COVID-19 response on women, especially pregnant women, would be enormous.

Our data align with other results showing a higher prevalence of income lost among women, especially in African regions, due to COVID-19-related factors.[24 25] Moreover, our study highlights the need for greater consideration of economic components of women's reproductive control, particularly during crisis periods. As we observed, many study participants were economically reliant on their partners or other external sources during the period of restrictions. Previous research indicates that women who have access to their own money use more FP and other SRH services.[26]

Also, we observed an attitudinal change towards FP services and altered intentions in use of such services among married and unmarried women due to the COVID-19 pandemic. Though the difference was not statistically significant, we found that more unmarried women than married women would have been very unhappy if they had gotten pregnant. A possible reason why unmarried women would not be happy if they had gotten pregnant could be the socioeconomic impact of COVID-19. During the COVID-19 pandemic, people were unsure whether or not they should get pregnant. Being pregnant is usually a happy time of anticipation and joyful planning. However, for many women, the COVID-19 pandemic clouded this period with fear, anxiety and uncertainty.[27] Presently, there is limited evidence available regarding mother-to-fetus transmission, transmission during delivery and transmission during breast feeding. Our findings showed that a significant proportion of unmarried women expected to wait for many years before giving birth due to the COVID-19 pandemic. This is in line with a study[28] that found more than one-third (37.3%) of their study participants who had intended to have a child prior to the pandemic changed their mind during the pandemic due to worries regarding future economic difficulties. The maternal and mental health implications during the COVID-19 pandemic have also been a concern among many pregnant women and studies showed higher rates of depression and self-harm ideation in women assessed during the COVID-19 outbreak than before.[29]

There are several limitations in this study. The cross-sectional design of this study limits the possibility to draw causal conclusions. Further, the use of secondary data limits control over the data collecting process and the ability to confirm information from participants. However, having the principal investigator of the primary study as part of this study further enabled us to understand the processes and gain clarity on every aspect of the primary study.

The study population included only women living in the Kinshasa province. Thus, the study findings cannot be generalised to women living in rural areas. Studies conducted before the COVID-19 pandemic have shown limited access and low uptake of FP services in rural communities.[30] It would have been interesting to include more representatives from rural settings and evaluate the effects of the restrictions on access to FP services in this group. Another aspect which can be regarded as a key strength, or a limitation is that the interview survey was phone based (which was necessary given the COVID-19 context). This further limited the ability of women in rural areas to be part of the study, due to the topography and limited access to phones among this group. Furthermore, our study cannot ascertain the long-term impact of the COVID-19 restrictions on FP. It would have been interesting to have other covariates, which could have

hindered access to services during the period in question and provide insights into possible future effects. As stated in the United Nations Population Fund's worldwide annual report: 'For every 3 months the lockdown continues, up to 2 million additional women may be unable to use modern contraceptives.'[31] Given the fact that many services were closed during the pandemic, this could have some psychological effects on women who wanted services, especially abortion services. However, our study did not take any such variable into consideration. Some strengths of this study include the data collection method used in the primary survey. It was well adapted to suit the current context and to limit infection rates for both data collectors and study participants. Also, the ODK tool kit aided proper processing and aggregation of data, minimising errors in the field. To further deal with bias and missing data in the study, we performed data weighting and imputation.

## CONCLUSION

While everyone has faced and is facing the unprecedented challenges of COVID-19, women are bearing the brunt of the economic and societal fallout of the pandemic. Effective support and interventions for women and girls during pandemics and crises are essential as they can have lasting beneficial effects on many aspects and domains of life. This study shows that the COVID-19 pandemic affected women in Kinshasa both socially and economically, with increased rates of economic reliance on a partner. Considering the potential disruptions to contraceptive services, healthcare providers could engage in discussions with women about self-administered subcutaneous depo shots or explore the use of long-term methods such as intrauterine devices to ensure consistent and reliable contraception. Most women did not attempt to access health services and some were afraid of being infected if they were to seek services.

**Author affiliations**
[1]Dalarna University, School of Health and Welfare, Falun, Sweden
[2]Department of Women's and Children's Health, Uppsala University, Uppsala, Sweden
[3]Kinshasa School of Public Health, University of Kinshasa, Kinshasa, Democratic Republic of Congo
[4]Department of Microbiology and Parasitology, Faculty of Science, University of Buea, Buea, Cameroon
[5]Department of Women's and Children's Health, Karolinska Institutet, Stockholm, Sweden

**Acknowledgements** The authors would like to thank the PMA programme for providing the data.

**Contributors** In crafting this manuscript, we, the authors, have worked collaboratively to ensure that each of us meets the criteria for authorship outlined by the International Committee of Medical Journal Editors (ICMJE). Our collective efforts have been focused on bringing together our unique expertise and perspectives to create a comprehensive and well-rounded piece of work. FNF conceptualised and drafted the manuscript. WU and EÅ contributed to the study development and supervised the study. PA (principal investigator) contributed to data and statistical screening. EJE contributed to data screening and statistical analysis. FNF and EJE contributed to the conceptual design of the analysis. FNF and EJE contributed to the interpretation and presentation of data. WU, PA and EÅ critically revised the entire manuscript. All authors have provided intellectual content and revised the manuscript.

**Funding** The authors have not declared a specific grant for this research from any funding agency in the public, commercial or not-for-profit sectors.

**Disclaimer** All the views and opinions presented in this paper are those of the authors and do not represent those of any institution.

**Competing interests** None declared.

**Patient and public involvement** Patients and/or the public were involved in the design, or conduct, or reporting, or dissemination plans of this research. Refer to the Methods section for further details.

**Patient consent for publication** Not applicable.

**Ethics approval** This study involves human participants and ethical approval was received from the University of Kinshasa School of Public Health (ESP/CEI/78/2020) and the John Hopkins Bloomberg School of Public Health Institutional Review Board. Participants gave informed consent to participate in the study before taking part.

**Provenance and peer review** Not commissioned; externally peer reviewed.

**Data availability statement** Data are available on reasonable request. Data may be obtained from a third party and are not publicly available.

**ORCID iD**
Falone Nkweleko Fankam http://orcid.org/0000-0001-7826-9382

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
