## [Reviewer comments · BMJ Open]

ARTICLE DETAILS

TITLE (PROVISIONAL)	The COVID-19 pandemic hits differently: examining its consequences for women's livelihoods and healthcare access. A cross-sectional study in Kinshasa DRC
AUTHORS	Nkweleko Fankam, Falone; Ugarte, Wiliam; Akilimali, Pierre; Ewane, Etah Junior; Eva, Åkerman

VERSION 1 – REVIEW

REVIEWER	Tang, Shangfeng School of medicine and health management, tongji mediical college, Huazhong university of science and techonolgy, Health mangament
REVIEW RETURNED	15-Apr-2023

GENERAL COMMENTS	1.The methods section of the abstract does not describe the sample selection method.2.Tables throughout the text do not use the three-line table format.3.The table design is faulty and should show the question items in one row.4.There is duplication of content in the text and graphs.5.The results section lacks specific statistical data to support.6.The format of the references is not standardized and unified.
---

REVIEWER	Agbemenu, Kafuli University at Buffalo School of Nursing
REVIEW RETURNED	10-Jun-2023

GENERAL COMMENTS	Thank you for this paper on the effects of Covid-19 on access to family planning services for women in DRC. It is a very needed contribution to the literature on family planning access in crisis situations. The introduction section was appropriate, using data from an already existing survey, with context from DRC, and international studies provided. Methodology: line 53: add '2020' after February. Patient and Public Involvement: it was somewhat confusing that public involvement was mentioned not to have happened, however, data is from the public. Data collection: line 22: did you mean to say administer SURVEYS using mobile phones? Line 26: "The instant aggregation of data allows for concurrent processing and course corrections while still in the field." This line is confusing. Please clarify. It is not clear to be how data is being instantly aggregated and processed. What does this mean? How is it being processed, and what is meant by "course corrections in the field"-were questions changed or refined in the field? Were data
--

	cleaned as received? Please clarify. Results: This section is adequately elucidated. Discussion: Ties in the intro and results sections appropriately. Regarding the recommendations, considering interruptions to contraceptives, you might suggest healthcare providers discussing self-administered sub-q depo shots, or long-term methods such as IUDS. The conclusion was appropriate in ending the paper.
--	--

VERSION 1 – AUTHOR RESPONSE

Reviewer 1

Dr. Shangfeng Tang, School of medicine and health management, tongji mediical college, Huazhong university of science and techonolgy

Comments to the Author:

1. *The methods section of the abstract does not describe the sample selection method.*

- **Response:** Thank you for this feedback. We have updated the methods section of the abstract to include a description of the sample selection method. Methods: We conducted a secondary analysis of performance monitoring for action (PMA) cross-sectional COVID-19 phone survey in Kinshasa DRC. The response rate was 74.7%. 1325 randomly selected women aged 15-49 from the Kinshasa-province who had previously participated in the PMA baseline survey participated in the survey. Bivariate and multivariate logistic regression was used to assess associations.

2. *Tables throughout the text do not use the three-line table format.*

- **Response:** We have ensured that tables throughout the text adhere to the three-line table format. A snapshot shown below:

Table 1: Demographic characteristics of the study population, N = 1325

Characteristics	Frequency (n)	Percentage (%)
Age group (years)		
15 – 24 (Youth)	474	35.8
25 – 34 (Young Adult)	464	35.0
35 – 49 (Adults)	387	29.2
Mean age (years)	29.49 ± 8.97	
Educational level ^a		
Primary	44	3.3
Secondary	884	66.8
Tertiary	395	29.9
Marital status ^a		
Married or cohabiting	620	47.0
Unmarried	703	53.0

3. *The table design is faulty and should show the question items in one row.*

- **Response:** We have made sure the tables display question items in a single row.

4. There is duplication of content in the text and graphs.

- **Response:** We have carefully reviewed the content and graphs to eliminate any duplication.

5. The results section lacks specific statistical data to support.

- **Response:** Thank you for raising this comment, even though the sentence seems incomplete for us to fully understand what you mean, we have gone through the section and tried to specify and highlighted statistical data to support the results section. If there are still comments to your questions kindly specify so we can clearly understand what you mean.

-

6. The format of the references is not standardized and unified.

- **Response:** We have standardized and unified the format of the references.

Reviewer 2

Dr. Kafuli Agbemenu, University at Buffalo School of Nursing

Comments to the Author:

Thank you for this paper on the effects of Covid-19 on access to family planning services for women in DRC. It is a very needed contribution to the literature on family planning access in crisis situations. The introduction section was appropriate, using data from an already existing survey, with context from DRC, and international studies provided.

Methodology: line 53: add '2020' after February.

- **Response:** We have added '2020' after February in the methodology section (line 53) to provide clearer temporal context.' survey between December 2019 and February 2020 ¹²

Patient and Public Involvement: it was somewhat confusing that public involvement was mentioned not to have happened, however, data is from the public.

- **Response:** Thank you for this feedback. We have addressed the confusion regarding public involvement and data collection. We have clarified that while public involvement did not occur in this secondary analysis research process, the data used in our study came from the public.

Data collection: line 22: did you mean to say administer SURVEYS using mobile phones?

- **Response:** We have revised the statement about data collection (line 22) to clarify to administer surveys. surveys were administered using mobile phones.

Line 26: "The instant aggregation of data allows for concurrent processing and course corrections while still in the field." This line is confusing. Please clarify. It is not clear to be how data is being instantly aggregated and processed. What does this mean? How is it being processed, and what is meant by "course corrections in the field"-were questions changed or refined in the field? Were data cleaned as received? Please clarify.

- **Response:** We have provided a more detailed explanation (line 26) of the instant aggregation of data, including concurrent processing and course corrections. 'The data collected during the study was instantly aggregated, enabling simultaneous processing and making it possible to make corrections and adjustments in real-time while still in the field Throughout the data collection stage, there was regular monitoring and feedback given to interviewers in the event of errors, missing data, or form submissions problems on the central server'. (Corrections like filling in a question where there need to be a skip based on the response of previous question).

Results: This section is adequately elucidated. Thank you

Discussion: Ties in the intro and results sections appropriately. Regarding the recommendations, considering interruptions to contraceptives, you might suggest healthcare providers discussing self-administered sub-q depo shots, or long-term methods such as IUDs.

- **Response:** We have considered your suggestion for recommendations in the discussion section, including the discussion of self-administered sub-q depo shots and long-term methods such as IUDs, in light of contraceptive interruptions.

The conclusion was appropriate in ending the paper. Thank you.

VERSION 2 – REVIEW

REVIEWER	Agbemenu, Kafuli University at Buffalo School of Nursing
REVIEW RETURNED	13-Aug-2023
GENERAL COMMENTS	Thank you for adequately addressing the reviewer's concerns. The manuscript has greater description and more effectively illustrates the experiences of the population.